# Inference- and Optimization-based Approximated Solver for Job-shop Scheduling Problem with Variable Set of Jobs

## Abstract

The Job-shop Scheduling Problem (JSP) is a well-known combinatorial optimization problem that arranges tasks for efficient processing. It is used in a broad range of industrial applications, such as smart manufacturing and transportation. **We focus on repeatedly solving scheduling instances with variable sets of jobs for real-world applications,** and we propose an inference-based model called JSPformer based on a data-driven scheme. **Our main contribution lies in enabling the training and inference of schedules for variable sets of jobs by encoding input data into job-wise feature vectors and utilizing a neural network for set-structured data.** Furthermore, for cases where a few minutes of additional computation is available, we propose JSPformer+Opt, a hybrid model of JSPformer and a local optimization. The local optimization is intended to make a more efficient schedule quickly from an inference solution. It uses part of the inference and optimizes the rest to improve the solution quality while reducing the problem size for fast computation. **In numerical experiments, we validated that JSPformer outperformed existing inference-based models and demonstrated its ability to handle instances with variable sets of jobs.**

## 1 Introduction

The Job-shop Scheduling Problem (JSP) is a well-known combinatorial optimization problem for determining the most efficient processing order of tasks in jobs. Tasks are assigned to machines, and a machine cannot process more than one task at the same time; consequently, we need to optimize the order of tasks for each machine. JSP has broad industrial applications, such as smart manufacturing and transportation. In these applications, schedules need to be modified according to a situation such as work delays and additional jobs. Furthermore, in such applications, the schedule needs to be updated before the deviation from actual operating status of machines increases. Therefore the optimization must be completed within a few minutes at most. For example, in manufacturing products, we can assume that a task is processed in a dozen minutes to an hour. Due to the randomness in job scheduling, including additional jobs and work delays, the schedule should be updated at intervals of a few minutes. Similar problems occur in parcel delivery services. When jobs are added over time, it is necessary to make a schedule under the constraints of the job release time in addition to the constraints considered in standard JSP. We call this problem *JSPRT (JSP with Release Time)*. It is an extension of the JSP, and reduces to the JSP when all jobs are released simultaneously. **This paper focuses on solving JSPRT within a given time limit after information about jobs arrives (Figure 1).**

Many approaches have been proposed to solve **such a scheduling problem as summarized in Figure 2, and classical algorithms struggle to overcome the trade-off between computation speed and solution accuracy. As a general-purpose exact method (Figure 2(i)), one can use Branch and Bound (B&B: Land & Doig (1960)) to solve the JSP, which iteratively updates the upper and lower bound. A more specific exact solution method for the JSP is CP-SAT solver**[1]**, which**

---

[1]A free CP-SAT solver is available through Google OR-tools:
https://developers.google.com/optimization/cp/cp_solver

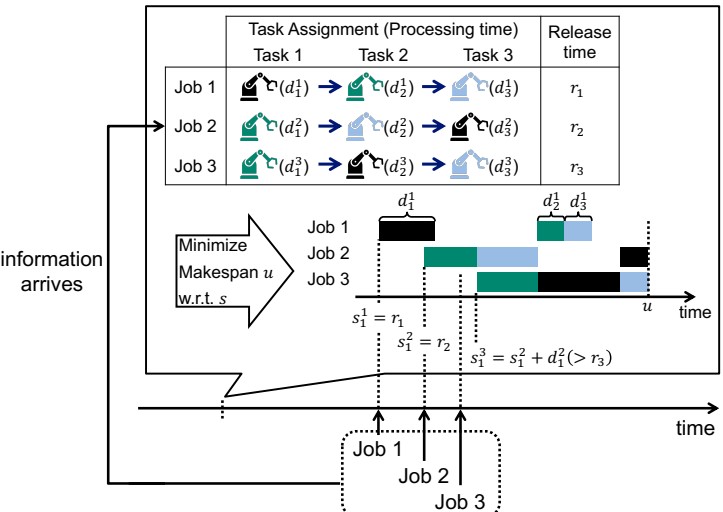

Figure 1: Illustration of a solver for **JSPRT** in job-variant settings. Each task is assigned to a machine visualized in the upper part. Colored rectangles represent tasks, where the color and width refer to the assigned machine and the processing time, respectively. To complete each job, three tasks must be processed in a specific order. The tasks are assigned to one of three machines at the same time, and they cannot overlap in time on a single machine. **After the job information arrives, we need to obtain a nearly-optimal schedule before the job arrival.**

**is based on constraint programming and SAT solver.** However, **such exact algorithms are** time-consuming for large-scale instances; **since JSP is an NP-hard problem, the computation time for obtaining an exact solution increases exponentially with the problem size**. Therefore, many other studies have focused on heuristic methods **(Figure 2(ii))** to obtain high-quality solutions in shorter time, such as rule-based methods (Dominic et al., 2004), local search, e.g. shifting bottleneck (Adams et al., 1988) or genetic algorithm (Lee et al., 1997). While most of these classical heuristics are intuitive and easy to understand, they tend to return sub-optimal solutions due to their simplicity.

**Considering scenarios of repeatedlly solving similar instances, it is promising to reuse the optimization process or the solution to an instance for other similar instances to obtain better solutions. Based on this idea, data-driven approaches have been explored to quickly infer high-quality solutions. They can be categorized into reinforcement learning and supervised learning. The reinforcement learning model (Figure 2(iii)), such as (Zhang et al., 2020), learns an appropriate dispatching rule using a dataset containing similar instances. However, it still produces sub-optimal solutions because it does not learn the solutions of these instances. When we can use datasets containing solutions of similar instances, supervised learning is expected to infer better quality solutions.** For example, several recent models, e.g., Kotary et al. (2021; 2022), can output high-quality solutions by capturing features of the given instances **and their solutions**. Notably, JSP-DNN (Kotary et al., 2022) **(Figure 2(iv))** reported that the nearly optimal solution of the JSP could be inferred from the processing times by learning solutions for similar instances with the same task assignment of all jobs. JSP-DNN can output a solution within one second by inference with a neural network and a post-processing algorithm to guarantee feasibility.

The disadvantage of such **supervised** learning-based models is that the architecture is designed for a *job-fixed setting*, a situation where the number of jobs and the task assignments of all jobs are fixed. The task assignment defines the order of the machines for each job by assigning tasks to the machines. **On the other hand, our aim is to solve JSPRT after the information about jobs arrives, and we cannot know the number of jobs and the task assignments.** Consequently, handling **JSPRT** in applications requires training and inference in *job-variant settings*, situations where the set of jobs varies **depending on the instances. In our assuming applications, the task assignments of all possible jobs are**

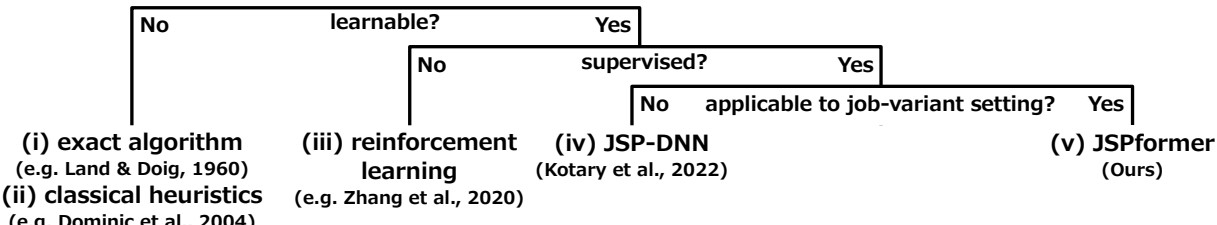

Figure 2: The methods for solving JSPRT are classified as the binary tree. Our model adopts a supervised learning approach to achieve a high-quality solution within a given time limit. Our model can handle job-variant settings by the capability of learning and inference on variable sets of jobs.

**known, but it is uncertain which jobs will arrive, requiring the ability to handle variable sets of jobs.**

To solve JSPRT within a given time limit in job-variant settings, we propose *JSPformer* **(Figure 2(v))**, a supervised learning-based model, **using solution datasets for similar instances.** JSPformer uses the Set Transformer (Lee et al., 2019), a state-of-the-art model for set-structured input data with variable length. To apply the Set Transformer to infer a solution of **JSPRT**, we need to modify the input data to reflect the task assignment, the training procedure, and the feasibility-recovery algorithm. Since the task assignment is represented as ordinal information, the Set Transformer cannot be applied directly to the input data. Accordingly, we **propose a method to** transform the input data into a set of job-wise vectors that reflects the ordinal information by a cumulative processing time. After training JSPformer, we can infer a high-quality solution for a variable number of jobs within a second; however, there is no guarantee that the inference solution satisfies all constraints in **JSPRT**. Similar to JSP-DNN, we use a feasibility-recovery algorithm to obtain a feasible solution from the inference.

Furthermore, as mentioned above, it is permitted to use a few minutes of computation time to search a better solution of **JSPRT** in our assumed applications. In order to make the best use of the given computation time limit, we also propose local optimization based on the JSPformer output, which we call JSPformer+Opt. After obtaining a schedule from JSPformer, JSPformer+Opt optimizes a simplified problem by fixing a portion of the inferred solution. Using this simplification, we can avoid an explosive increase in computation time and apply an exact algorithm since the computation time of the exact algorithm increases exponentially with the number of jobs.

We evaluate our model through two experimental settings; namely, job-fixed and job-variant settings. In contrast to job-variant setting, the task assignment is fixed in the job-fixed setting, and only the processing time and the release time vary. By comparing our model with JSP-DNN or heuristics, we experimentally showed that our model works well in both settings. Moreover, JSPformer+Opt produced better or more competitive solutions for **JSPRT** instances within a minute compared to optimized solutions using an exact solver for over 30 minutes.

## 2  Formulation of **JSPRT**

In this section, we formulate the **JSPRT** to optimize the schedule for a given set of jobs. In the **JSPRT**, we consider $J$ jobs to be processed by $M$ machines, where each job $j$ consists of $T$ tasks and each task can only be performed on a specific machine. According to the classical setting, we assume that the number of tasks in each job is equal to the number of machines ($T = M$) and that each job uses all machines once. If a job has fewer than $M$ tasks, we can use a dummy task whose processing time is zero. In this way, we can assume that the number of tasks $T$ for each job is fixed at $M$, and only the number of jobs $J$ varies. It is also assumed that the processing time is given a priori. The objective is to find a schedule that minimizes the latest completion time of all jobs, namely *makespan*. The **JSPRT** is formulated as a model that considers three types of constraints:

(c1) task-precedence constraint: Tasks in a job have a fixed processing order; $p + 1$-th task processing cannot be started until the $p$-th task processing is completed.

(c2) no-overlap constraint: no machine can process more than one job at a time.

(c3) release-time constraint: job $j$ cannot be processed before its release time $r_j$.

Different from JSP, which considers only constraints (c1) and (c2), job-variant settings needs to consider the fact that the number of jobs varies with time, which is reflected in the release-time constraint (c3). Let $\sigma_p^j \in \{1, \cdots, M\}$ be the machine that processes the $p$-th task of job $j$, $d_p^j$ be its processing time, and $r_j$ be the release time of job $j$. For simplicity, we denote by $\boldsymbol{\sigma}$ the $J \times M$ matrix with $(j, p)$ component as $\sigma_p^j$, by $\boldsymbol{d}$ the $J \times M$ matrix with $(j, p)$ component as $d_p^j$, and by $\boldsymbol{r}$ the $J$-dimensional vector with the $j$-th component as $r_j$. Accordingly, the **JSPRT** is formulated as

$$(P) \qquad \underset{\boldsymbol{s}}{\text{minimize}} \quad u = \max_j \left( s_T^j + d_T^j \right)$$

$$\text{subject to} \quad s_{p+1}^j \geq s_p^j + d_p^j \ (\forall p, j) \qquad\qquad \text{(c1)}$$

$$s_p^j + d_p^j \leq s_{p'}^{j'} \text{ or } s_{p'}^{j'} + d_{p'}^{j'} \leq s_p^j \qquad\qquad \text{(c2)}$$

$$(\forall (j, p, j', p') \text{ s.t. } \sigma_p^j = \sigma_{p'}^{j'})$$

$$s_1^j \geq r_j \ (\forall j) \qquad\qquad \text{(c3)}$$

where $s_p^j$ and $u$ are variables that represent the task start times of the $p$-th process of job $j$ and the makespan, respectively. The notations are summarized in Table 1 **and (P) can be re-formulated as a mixed integer optimization problem (see Appendix A)**. Our goal is to obtain $\boldsymbol{s} \in \mathbb{R}^{J \times M}$ that minimizes $u$ from the input data consisting of the task assignment $\boldsymbol{\sigma}$, the processing time of tasks $\boldsymbol{d} \in \mathbb{R}^{J \times M}$, and the release time of jobs $\boldsymbol{r} \in \mathbb{R}^J$, where $\boldsymbol{s}$ denotes the $J \times M$ matrix with the $(j, p)$ component as $s_p^j$. This problem has a combinatorial structure mainly due to the no-overlap constraint (c2); in the optimization process, it is necessary to choose which constraints in (c2) should be satisfied. Figure 1 depicts an example of a **JSPRT** instance with two or three jobs and three machines. The schedule $\boldsymbol{s}$ can be visualized through the Gantt chart at the bottom of Figure 1 and we can see that this schedule satisfies the no-overlap constraint (c2).

## 3 Related work

This section summarizes related work on the JSP. We describe classical approaches in Section 3.1 and then mention more recent machine learning-based approaches in Section 3.2.

### 3.1 Classical Approaches

Since the JSP can be formulated as a mixed integer optimization problem, it is possible to use Branch and Bound (B&B) (Land & Doig, 1960), a well-known exact algorithm for combinatorial

Table 1: Notations

| | |
|---|---|
| $J$ | number of jobs |
| $j \in \{1, \cdots, J\}$ | job index |
| $T$ | number of tasks per job |
| $M$ | number of machines (generally $T = M$) |
| $p \in \{1, \cdots, T\}$ | task index |
| $\sigma_p^j \in \{1, \cdots, M\}$ | task assignment of $p$-th task of job $j$ |
| $d_p^j \geq 0$ | processing time of $p$-th task of job $j$ |
| $r_j$ | release time of job $j$ |
| $s_p^j \geq 0$ | start time of $p$-th task of job $j$ |
| $u \geq 0$ | makespan |

optimization problems including scheduling problems (Brucker et al., 1994; Peterkofsky & Daganzo, 1990; D'ariano et al., 2007; Brucker et al., 1998). Due to its wide applicability, B&B is at the core of combinatorial optimization solvers such as CBC[2], CPLEX[3], and Gurobi[4] and can be applied to other combinatorial optimization problems, such as the traveling salesman problem (Balas & Toth, 1983), the vehicle routing

---

[2]https://github.com/coin-or/Cbc
[3]https://www.ibm.com/products/ilog-cplex-optimization-studio
[4]https://www.gurobi.com

problem (Lysgaard et al., 2004), and the bin-packing problem (Valério de Carvalho, 1999). **As another exact algorithm for the JSP, CP-SAT (Ohrimenko et al., 2009) solves the problem by reducing the search space through deriving new constraints from several constraints.** These exact algorithms can work as useful solutions when the problem size is small enough for the given computation time, but it is not suitable for obtaining good solutions to large-scale problems.

For fast approximation, we can use heuristic approaches. The simplest way is to use dispatching rules that determine a processing order (e.g., (Dominic et al., 2004)). Although there are several types of dispatching rules such as shortest processing time or least work remaining, these rules basically output a schedule with much lower efficiency than one by an exact solution method. According to previous work (Zhang et al., 2020; Kotary et al., 2022), a rule-based schedule is more than 20% worse than a schedule made using an exact algorithm. As another heuristic, local search algorithms aim to improve such a sub-optimal solution by altering the solution locally. As in the dispatching rules, there are several local search algorithms, such as shifting-bottleneck methods (Adams et al., 1988), genetic algorithms (Lee et al., 1997), **large neighborhood search (Godard et al., 2005). In cases where the local search algorithms cannot improve the solution, failure-directed search (Vilím et al., 2015) is proposed by to avoid unnecessary exploration of the search space.** As mentioned in Section 1, these algorithms do not use an implicit pattern in the dataset, so they are inefficient for solving similar problems repeatedly.

### 3.2 Machine Learning-based Approaches

Unlike classical methods, some recent studies apply machine learning models for combinatorial optimization problems by capturing a pattern in the optimal solutions contained in a dataset. For example, a supervised learning model has been proposed to solve general mixed integer quadratic optimization problems by predicting tight constraints and discrete variables (Bertsimas & Stellato, 2022). Another approach is to construct an approximated solver by jointly training a prediction and an optimization models (Wilder, 2019; Mandi et al., 2022). **Another direction for integrating machine learning and optimization is decision-focused learning(e.g. (Mandi et al., 2023)), where the input parameters of an optimization problem are uncertain and machine learning model predicts the parameters to minimize the effect of uncertainty; however, this setting is out of the scope of this paper and we focus on learning about the optimization process itself under the input parameters are known.**

Machine learning methods specified for JSP have also been studied. One method (Zhang et al., 2020) tried to find an appropriate dispatching rule from a solution dataset of the JSP, enabling more efficient scheduling than other rule-based scheduling. Similarly, a different approach (Ingimundardottir & Runarsson, 2018) used imitation learning to learn an efficient dispatching rule for the JSP. **Also, there are several models based on reinforcement learning (Song et al., 2023; Chen et al., 2022) to handle dynamic events such as machine breakdown or work delays. Furthermore, there are machine learning-based studies for other types of scheduling problems such as resource-constrained scheduling (Mao et al., 2016; Teichteil-Königsbuch et al., 2023; Chen & Tian, 2019).**

While these models learn dispatching rules instead of the optimal schedule itself, JSP-DNN (Kotary et al., 2022) learns the optimal schedules and the problem constraints directly by deep neural networks in the job-fixed setting. The numerical experiments in the paper have shown that in some cases, JSP-DNN outputs a high-quality solution as well as a 30-minute application of an exact algorithm while, on the other hand, rule-based algorithms output a worse solution than a 1-minute application of an exact algorithm. This comparison demonstrates the solution quality of JSP-DNN. JSP-DNN learns the solutions for JSP in the job-fixed setting with $M$ machines and $J$ jobs by preparing three types of neural networks: $M$ machine-wise neural networks, $J$ job-wise neural networks, and a shared layer (Figure 3(a)). The machine(job)-wise network encodes $J(M)$-dimensional vectors of processing times corresponding to the machine(job) into a feature vector, and the shared layer infers a solution from these features. Each layer is implemented as a two-layer perceptron and the input dimension of the machine-wise networks and the number of job-wise networks are fixed to $J$. In the training procedure, the task assignment $\boldsymbol{\sigma}$ is fixed and implicitly learned by job-wise networks; the $j$-th job-wise network for the $j$-th job is trained for the corresponding task assignment $(\sigma_j^1, \cdots, \sigma_j^M)$. Since the inferred solution may not satisfy some of the constraints, the paper also proposed a post-processing algorithm to recover a feasible solution from the inference. The feasibility-recovery algorithm

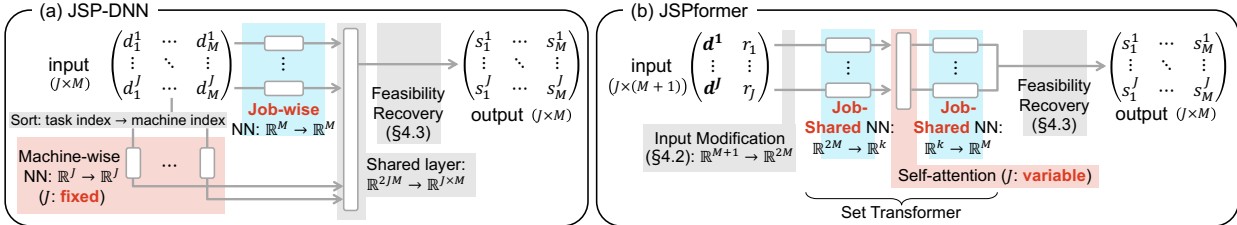

Figure 3: Architecture of JSP-DNN (a) and our model, JSPformer (b). $J$ and $M$ denote the number of jobs and that of machines, respectively. Since JSP-DNN holds job-wise neural networks, JSP-DNN cannot be used for **JSPRT** with a variable number of jobs $J$. To tackle this problem, JSPformer adopts Set Transformer, which enables training and inference with a variable set of jobs. JSPformer focuses on the **JSPRT** and release times $r_1, \cdots, r_J$ are added to the input.

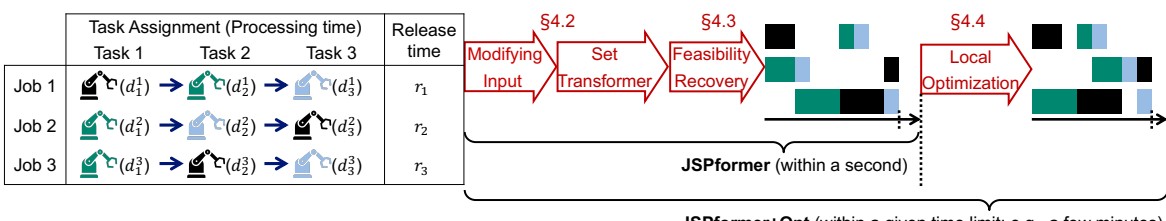

Figure 4: Procedures of JSPformer and JSPformer+Opt, with a **JSPRT** instance having three jobs and three machines. Given the input data, JSPformer infers a feasible solution through an input modification (Section 4.2), an inference with Set Transformer (Section 4.2), and a feasibility-recovery algorithm (Section 4.3). JSPformer+Opt improves the solution of JSPformer by local optimization (Section 4.4). JSPformer can infer a solution within a second, whereas JSPformer+Opt improves the solution quality in a short time (e.g. a few minutes), and thus we can choose either of these two models according to the application.

uses the inference to define an order of tasks by comparing the inferred task start time (Kotary et al., 2021) or the middle value between the inferred task start and end times (Kotary et al., 2022). This order determines which constraints of (c2) are satisfied, thus making the problem much easier. By combining the inference and the feasibility-recovery algorithm, JSP-DNN can output a solution that is much closer to the optimal solution than a rule-based solution. From the viewpoints of computational speed and accuracy, our study uses JSP-DNN, a supervised learning model for JSP, as the baseline.

While JSP-DNN performs well in the job-fixed setting, it faces limitations when applied to job-variant settings. JSP-DNN assumes that the task assignment $\boldsymbol{\sigma}$ is fixed since JSP-DNN has a job-wise neural network (Figure 3(a)) that implicitly reflects $\boldsymbol{\sigma}$. This is not applicable in job-variant settings, such that $\boldsymbol{\sigma}$ varies with the set of jobs. If we applied JSP-DNN to job-variant settings, we needed to determine the set of jobs in order to prepare job-wise networks in advance. For example, considering three jobs with different task assignments, JSP-DNN needs three job-wise networks for each job, even if each instance in the dataset has two jobs. One could assume a set of all possible jobs $\mathcal{J}_{ub}$, and employ JSP-DNN for a job-variant setting with the fixed task assignment $\boldsymbol{\sigma}$ consisting all task assignments of jobs in $\mathcal{J}_{ub}$. However, there would be many redundant and inefficient cases where the actual number of jobs ends up being much smaller than $|\mathcal{J}_{ub}|$. Thus, JSP-DNN cannot be applicable for job-variant settings. In addition, JSP-DNN can infer a high-quality solution within about a second, while we are able to spend a few minutes in our assumed applications, introduced in Section 1. This means that we have time to improve the solution quality from the inference.

# 4 Proposed method

In this section, we propose JSPformer and JSPformer+Opt to address the following two issues: (i1) constructing an inference model for a variable number of jobs and (i2) improving the solution quality. JSPformer is a data-driven model for the **JSPRT** that addresses the first issue (i1) by encoding the input data $(\boldsymbol{d}, \boldsymbol{r}, \boldsymbol{\sigma})$ into a set of job-wise feature vectors and using a neural network for set-structured data (to check the architecture, see Figure 3(b)). The second issue (i2) is addressed by JSPformer+Opt, a hybrid model of JSPformer and the local optimization. The local optimization improves the solution quality by optimizing a portion of the inference solution. Depending on the given time available, the problem size can be adjusted in the local optimization. In the following section, we give an overview of the proposed methods.

## 4.1 Overview

Figure 4 gives an overview of the proposed models. JSPformer focuses on rapid computation that combines a neural network for solution inference and a feasibility-recovery algorithm. JSPformer basically follows the same training procedure as JSP-DNN (detailed in Appendix C). Moreover, we propose JSPformer+Opt, a hybrid model of JSPformer and local optimization, under the assumption that we have more than a few seconds to spend on the optimization. From the next subsection, we give details on the JSPformer inference model (Section 4.2), an algorithm to recover feasibility (Section 4.3), and the local optimization method (Section 4.4).

## 4.2 Inference with Set Transformer from Modified Input Data

JSPformer addresses the first issue (i1) by regarding input $(\boldsymbol{d}, \boldsymbol{r}, \boldsymbol{\sigma})$ and output $\boldsymbol{s}$ as job-wise set-structured data. In this way, we can use a neural network model for set-structured data that support variable-size inputs. As an implementation, we adopted Set Transformer (Lee et al., 2019), a state-of-the-art model for set-structured data.

Given a set of processing times $\boldsymbol{d} \in \mathbb{R}^{J \times M}$, a job-release time $\boldsymbol{r} \in \mathbb{R}^M$, and a task assignment for jobs $\boldsymbol{\sigma} \in \{1, \cdots, M\}^{J \times M}$, we regard $(\boldsymbol{d}, \boldsymbol{r})$ as a set of job-wise input data $D := \left\{ \left( \boldsymbol{d}^j, r_j \right) \right\}_{j=1}^J$, and JSPformer first infers job-wise start time $\left\{ \boldsymbol{s}^j \right\}_{j=1}^J$ with a neural network $f_{\boldsymbol{\theta}}$ parametrized by $\boldsymbol{\theta}$, where $\boldsymbol{d}^j = (d_1^j, \cdots, d_M^j) \in \mathbb{R}^M$ and $\boldsymbol{s}^j = (s_1^j, \cdots, s_M^j) \in \mathbb{R}^M$ denote the vectors of the processing time and the start time of the job $j$, respectively. As explained in Section 3, we cannot use the same architecture as JSP-DNN for job-variant settings. Consequently, we instead use Set Transformer, a model designed to handle set-structured variable-size input data consisting of self-attention and row-wise transformation. Self-attention can be computed regardless of the number of input vectors, making it applicable to our targeted **JSPRT**. Intuitively, the self-attention reflects the no-overlap constraint (c2), while the row-wise transformation reflects the task-precedence constraint (c1) and the release-time constraint (c3).

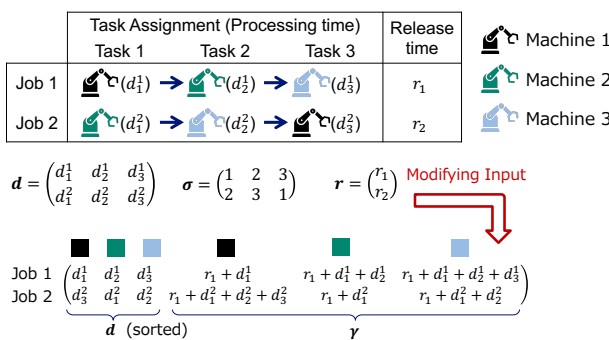

Figure 5: Detailed process used to modify inputs for the solution inference with a variable number of jobs. Modified input data reflect the task assignment. We use the matrix at the bottom of this figure which is indexed by the job index and the machine index (black → green → blue) instead of the task index.

Since $D$ has no information on task assignment $\boldsymbol{\sigma}$, we need to modify the input data; given the input $D$, we constructed a cumulative processing time $\boldsymbol{\gamma} = \left\{ \boldsymbol{\gamma}^j \right\}_{j=1}^J$, defined as $\gamma_1^j = r_j + d_1^j$ and $\gamma_{p+1}^j = \gamma_p^j + d_p^j$, and reordered $\boldsymbol{\gamma}^j$ according to the machine index. Since inequality $\gamma_{p+1}^j > \gamma_p^j$ reflects the task sequence of each job, we can input the task assignments to machines by sorting $\boldsymbol{\gamma}^j \in \mathbb{R}^M$ and $\boldsymbol{d}^j \in \mathbb{R}^M$ according

to the machine indices. Additionally, $\boldsymbol{\gamma}$ has release-time information by $\gamma_1^j$. Figure 5 shows an example of the modification; in this example, the input data reflects the task assignments by the column indices. Set Transformer can learn the task-precedence constraints (c1) and release-time constraints (c3) in a row-wise transformation. Our input modification may be interpreted as a proposition for positional encoding in Transformer architecture for **JSPRT**; just as the positional encoding represents an order of sequential data, the modified input $\boldsymbol{\gamma}$ represents the processing order derived from the task-precedence constraint (c1).

### 4.3 Feasibility Recovery

Although JSPformer is trained to satisfy constraints (c1)-(c3), there is no guarantee that the inference result $\hat{\boldsymbol{s}}$ satisfies these constraints. We used a procedure similar to JSP-DNN to recover a feasible solution from the inference. **Although the original algorithm does not support the release-time constraint (c3), the modified version of it supports (c3)** and if $r_j = 0$ for every job $j$, the feasibility-recovery algorithm is the same as that of JSP-DNN. By using $\hat{\boldsymbol{s}}$, we could define the order of tasks $\leq_{\hat{s}}$ as $\hat{s}_j^p \leq \hat{s}_{j'}^{p'} \iff (j,p) \leq_{\hat{s}} (j',p')$ for two tasks $(j,p), (j',p')$ that share the same machine. Since this order only reflects the no-overlap constraint (c2), it may contradict the task-precedence constraint (c1). If not, we can obtain a feasible solution of (P) by optimizing the following problem.

$$(P') \qquad \underset{\boldsymbol{s}}{\text{minimize}} \quad u = \max_j (s_T^j + d_T^j)$$

$$\text{subject to} \quad s_{p+1}^j \geq s_p^j + d_p^j \ (\forall p) \tag{c1}$$

$$s_p^j + d_p^j \leq s_{p'}^{j'} \tag{c2'}$$
$$(\forall (j,p), (j',p') \text{ s.t. } (j,p) \leq_{\hat{s}} (j',p'))$$

$$s_1^j \geq r_j \ (\forall j) \tag{c3}$$

This problem (P') belongs to the linear optimization problems and can be optimized in a much shorter time than the original problem (P) (e.g., within a second for 20 jobs). In the case where $\leq_{\hat{s}}$ contradicts the task-precedence constraint (c1), we use a greedy algorithm in the same way as JSP-DNN (detailed in Algorithm 2 in Appendix B), which gives a feasible solution to (P) by reconstructing an order $\leq_{\hat{s}}$ based on the inference $\hat{\boldsymbol{s}}$.

### 4.4 Local Optimization from Inference Solution

We can obtain a high-quality feasible solution by recovering feasibility using the order of tasks from the inference. After that, we can further refine this solution by local optimization defined below. In the local optimization, we divided the set of jobs into inference-based jobs and optimization-based jobs and then fixed the order of tasks for the inference-based jobs. In this way, we could reduce the size of problem (P) significantly, and the solution could be refined by applying the solver over a few minutes. Given an order $\leq_{\hat{s}}$ between the inference-based jobs and the set of optimization-based jobs $\mathcal{J}_o$, we could formulate a simplified version of (P) by replacing (c2) with (c2') for any two tasks in the inference-based jobs. This simplification reduces the problem complexity; the original problem (P) has at most $(J!)^M$ patterns, which is reduced to $(J!/(J-K)!)^M$ in the simplified problem, where $K = |\mathcal{J}_o|$.

## 5 Experiments

To verify the effectiveness of the proposed method, we evaluated it in two experimental settings. The first setting is inherited from JSP-DNN; namely, the neural network is trained in the job-fixed setting. The aim of this experiment was to investigate differences between JSPformer(+Opt) and JSP-DNN(+Opt), where JSP-DNN+Opt refers to a hybrid model of JSP-DNN and local optimization for fair evaluation with JSPformer+Opt. **To compare our model to other methods in Figure 2, we also evaluated (i) an exact solver, (ii) a classical heuristic, and (iii) a reinforcement learning-based model.**

In the second setting, an experiment shows that our model is applicable to job-variant settings, as defined at the beginning of Section 1. In the second experiment, we randomly removed a part of jobs from the

original instances in the dataset construction. In this way, we could check whether JSPformer was able to handle a variable number of jobs while maintaining **that JSPformer knows the task assignments of all possible jobs**.

## 5.1 Data Generation

For experiments in both job-fixed and job-variant settings, we prepared three types of datasets based on three JSP benchmark instances from JSPLIB[5], namely, *La21* with 15 jobs and 10 machines, *La16* with 10 jobs and 10 machines, and *La11* with 20 jobs and 5 machines.

In the job-fixed setting, we used these instances to prepare the datasets, as in the experiment of JSP-DNN (Kotary et al., 2022). For all instances in each dataset, the number of jobs and the number of machines are the same as in the original instance. For each instance, we generated 1,500 individual instances by slowing down the processing time; as in the original experiment of JSP-DNN (Kotary et al., 2022), the processing time was randomly slowed down up to 50% from the original processing time. The instances were divided into 1,490 training data and 10 test data.

Conversely, in job-variant settings, a dataset containing instances with different sets of jobs is required. To construct such a dataset, we created a dataset consisting of variable-size instances by randomly removing jobs from La21, La16, and La11 instances. As in the job-fixed setting, we randomly varied the job-release time and processing time. For each instance, we prepared a total of 3,000 instances with up to 3 jobs removed. For example, the La21 dataset for job-variant settings contains instances with 12-15 jobs. In the dataset creation, we uniformly choose the number of jobs removed. For each of the three datasets and each number of removed jobs, we prepared 10 test data, and trained JSPformer on the remaining 2,960 training data.

It should be emphasized that, unlike datasets for images or audio, the preparation of training data requires computation time to run an exact algorithm for optimization. Following the experimental settings in the study of JSP-DNN, we set a time limit of 1,800 seconds for each instance and used the best solutions within the time limits, resulting in 93.75 days of computation time **for the dataset generation of the job-fixed setting.** (3 types of datasets × 1,500 instances × 1,800 seconds). The original instances from JSPLIB do not have the release time constraint (c3), so we randomly generated the release times. We sorted the generated release time according to the job index in order to give regularity to the release times as well as to the processing time. As an exact algorithm, we used **B&B solver. Specifically, we used** CBC; CBC is widely used as a default solver, such as in PuLP (Python modeling library for optimization) or Google OR-tools. Due to the time limit, the datasets contained sub-optimal solutions; consequently, there is the possibility that JSP-DNN and JSPformer output better solutions than the dataset solutions. We prepared the three datasets for both job-fixed and job-variant settings.

## 5.2 Training Details

In the job-fixed setting, we used an inference model for each dataset, as in the experimental setup of JSP-DNN.

While JSPformer can handle both job-fixed and job-variant settings, the vanilla JSP-DNN is only available in the job-fixed setting **because JSP-DNN has job-wise neural networks**. To use JSP-DNN as a baseline for JSPformer, we modified JSP-DNN to address this problem by using the input modification; we used the modified input data $(\boldsymbol{d}, \boldsymbol{\gamma})$ as the input of JSP-DNN to reflect the job-release time.

The training and evaluation processes were conducted using an Apple M2 MacBook Pro with 16 GB RAM. We report the best results after tuning several hyperparameters such as the learning rate (detailed in Appendix D).

---

[5]https://github.com/tamy0612/JSPLIB

### 5.3 Experimental Settings

Since we aim to update the schedule within a period of a few minutes, we set one minute as the time limit for local optimization. In the local optimization, the number of optimization-based jobs $K$ should be adjusted to this computation time limit. According to the estimated problem complexity $(J!/(J - K)!)^M$, we use relatively small $K$ when $J$ or $M$ is large (Table 2, 3). To ensure that computation time minimally affected processing, we chose $K$ jobs with late release times as the optimization-based jobs in the experiment.

### 5.4 Evaluation Metrics

In the experiments, we used the following four metrics to measure inference and optimization performance. Here, ↑ or ↓ indicates that a higher or lower value is more favorable, respectively.

• Prediction Error ↓: To measure how well the inference models fit the datasets, the columns for prediction error report the L2 distance between the output of the inference $\hat{s}$ and the target solution $s$ obtained by CBC solver.

• Constraint Violation ↓: To measure the magnitude of the constraint violations (c1)-(c3), the columns for constraint violation report the L2 distance between the output of the inference before the feasibility recovery $\hat{s}$ and the solution after the feasibility recovery $\tilde{s}$.

• Gap ↓: Gap measures the gap of objective values between the two solutions, one obtained by the CBC solver and the other obtained by the inference model or the hybrid model. Since the CBC solver has a time limit of 1,800 seconds as mentioned in Section 5.1, the solutions by the CBC solver are not always optimal. In some cases, the inference solution is better than the solution by the CBC solver, in which case the gap becomes negative.

• #Better Solutions ↑: As mentioned above, the inference model or the hybrid model may output a better solution than that by the CBC solver. We count such instances and report them in the column of #better solutions.

### 5.5 Experimental Results for Job-fixed Setting

Table 2 shows the numerical results of **(i) CBC, an exact solver**, (ii) MWR+SB, a classical heuristics consisted of dispatching rule (Most Work Remained: MWR) (Dominic et al., 2004) and shifting bottleneck (SB) (Adams et al., 1988), (iii) L2D(+Opt), a reinforcement learning-based model (Zhang et al., 2020), (iv) JSP-DNN(+Opt), and (v) JSPformer(+Opt). In Appendix E, we report numerical results with other several dispatching rules (SPT, LWR, LOR, and MOR) and shifting bottleneck. For a fair comparison, we report the results of JSP-DNN+Opt **and L2D+Opt, hybrid models of JSP-DNN and L2D** with 1-minute local optimization (computation time discussed in Section 6.1). From this table, we can see that JSPformer+Opt produced the best results compared to the other models. Especially, classical heuristics output sub-optimal solutions, as mentioned in Section 1. Comparing the results for inference ($K = 0$) of JSPformer and JSP-DNN, JSPformer inferred more accurate solutions (prediction error and constraint violation) and performed as well as or better than JSP-DNN (gap). **Intuitively, the difference is due to the capability to handle multiple constraints (c1)-(c3); JSP-DNN is built on the assumption that all jobs are released simultaneously, which results in less accurate results compared to our proposed method. In contrast, JSPformer is specifically designed to handle scenarios where jobs have release times. Further analysis of prediction error and constraint violation in Appendix F revealed that JSP-DNN struggled particularly with satisfying both the release time and no-overlap constraints simultaneously.**

In combination with local optimization, we obtained an even better solution. Except for La16, JSPformer+Opt outperformed the 30-minute applications of the CBC solver.

**Regarding the comparison between JSPformer and L2D, a possible factor for the difference between L2D and JSPformer could be the use of the training dataset. While JSP-DNN and JSPformer learn from the solutions in the training dataset, L2D does not use these solutions.**

Table 2: Experimental Results of **(i) exact solver with 1-minute time limit (CBC (1min.)), (ii) classical heuristics (MWR+SB), (iii) reinforcement learning-based model (L2D(+Opt)), (iv) JSP-DNN(+Opt), and (v) our model, JSPformer(+Opt) in the job-fixed setting. For a fair comparison, a hybrid model of L2D(JSP-DNN) and the local optimization is also presented as L2D+Opt(JSP-DNN+Opt), respectively.** $K$ denotes the number of jobs to optimize after the inference and $K = 0$ refers to the results of inference solutions.

| base instance | model | $K$ | Prediction Error ↓ | Constraint Violation ↓ | Gap ↓ | # Better Sol. ↑ |
|---|---|---|---|---|---|---|
| La21 | (i) CBC (1 min.) | - | - | - | 0.042±0.049 | 3/10 |
| ($J$=15, $M$=10) | (ii) MWR+SB | - | - | - | 0.367±0.086 | 0/10 |
| | (iii) L2D | 0 | - | - | 0.043±0.053 | 1/10 |
| | | 2 | - | - | 0.008±0.064 | 6/10 |
| | L2D+Opt | 3 | - | - | -0.028±0.057 | 9/10 |
| | | 4 | - | - | -0.032±0.035 | 8/10 |
| | (iv) JSP-DNN | 0 | 2,028 | 843 | -0.012±0.026 | 9/10 |
| | | 2 | - | - | -0.058±0.038 | 9/10 |
| | JSP-DNN+Opt | 3 | - | - | -0.051±0.037 | 8/10 |
| | | 4 | - | - | -0.024±0.031 | 7/10 |
| | (v) JSPformer | 0 | 1,971 | 873 | -0.033±0.037 | 9/10 |
| | | 2 | - | - | -0.067±0.044 | 9/10 |
| | JSPformer+Opt | 3 | - | - | **-0.069±0.036** | 9/10 |
| | | 4 | - | - | -0.041±0.029 | 9/10 |
| La16 | CBC (1 min.) | - | - | - | 0.051±0.032 | 0/10 |
| ($J$=10, $M$=10) | (ii) MWR+SB | - | - | - | 0.299±0.080 | 0/10 |
| | (iii) L2D | 0 | - | - | 0.180±0.047 | 0/10 |
| | | 2 | - | - | 0.102±0.030 | 0/10 |
| | L2D+Opt | 4 | - | - | 0.072±0.032 | 0/10 |
| | | 6 | - | - | 0.049±0.024 | 0/10 |
| | (iv) JSP-DNN | 0 | 1,243 | 691 | 0.077±0.017 | 0/10 |
| | | 2 | - | - | 0.071±0.014 | 1/10 |
| | JSP-DNN+Opt | 4 | - | - | 0.059±0.022 | 1/10 |
| | | 6 | - | - | 0.032±0.014 | 0/10 |
| | (v) JSPformer | 0 | 1,248 | 454 | 0.053±0.014 | 0/10 |
| | | 2 | - | - | 0.037±0.010 | 0/10 |
| | JSPformer+Opt | 4 | - | - | **0.016±0.014** | 0/10 |
| | | 6 | - | - | 0.049±0.028 | 0/10 |
| La11 | CBC (1 min.) | - | - | - | 0.121±0.088 | 2/10 |
| ($J$=20, $M$=5) | (ii) MWR+SB | - | - | - | 0.128±0.107 | 1/10 |
| | (iii) L2D | 0 | - | - | -0.058±0.029 | 10/10 |
| | | 2 | - | - | -0.080±0.029 | 10/10 |
| | L2D+Opt | 4 | - | - | -0.070±0.029 | 10/10 |
| | | 6 | - | - | -0.041±0.035 | 9/10 |
| | (iv) JSP-DNN | 0 | 2,519 | 1,371 | -0.097±0.035 | 10/10 |
| | | 2 | - | - | -0.102±0.035 | 10/10 |
| | JSP-DNN+Opt | 4 | - | - | -0.094±0.032 | 10/10 |
| | | 6 | - | - | -0.080±0.035 | 10/10 |
| | (v) JSPformer | 0 | 2,502 | 1,362 | -0.082±0.053 | 9/10 |
| | | 2 | - | - | **-0.106±0.036** | 10/10 |
| | JSPformer+Opt | 4 | - | - | -0.094±0.037 | 10/10 |
| | | 6 | - | - | -0.056±0.056 | 7/10 |

Table 3: Experimental Results of JSPformer and JSPformer+Opt in job-variant settings. $K$ denotes the number of jobs to optimize after the inference and $K = 0$ refers to the results of JSPformer. $L$ denotes the number of removed jobs. As a baseline, results of a heuristic method (MWR+SB) are also reported.

| base instance | $K$ / model | Gap(total) | Gap($L=0$) ↓ | Gap($L=1$) ↓ | Gap($L=2$) ↓ | Gap($L=3$) ↓ | # Better Sol. ↑ |
|---|---|---|---|---|---|---|---|
| La21 ($J$=15, $M$=10) | 0 | 0.069±0.068 | 0.003±0.048 | 0.084±0.062 | 0.072±0.034 | 0.118±0.065 | 5/40 |
| | 2 | 0.007±0.061 | -0.035±0.061 | 0.021±0.068 | 0.002±0.027 | 0.039±0.052 | 15/40 |
| | 3 | **-0.014±0.052** | **-0.042±0.060** | **-0.016±0.049** | **-0.011±0.029** | 0.014±0.046 | **26/40** |
| | 4 | 0.013±0.062 | 0.006±0.064 | 0.028±0.076 | 0.022±0.043 | **-0.003±0.052** | 20/40 |
| | MWR+SB | 0.323±0.131 | 0.356±0.168 | 0.362±0.107 | 0.287±0.069 | 0.288±0.137 | 0/40 |
| La16 ($J$=10, $M$=10) | 0 | 0.136±0.065 | 0.176±0.051 | 0.135±0.048 | 0.155±0.061 | 0.078±0.052 | 0/40 |
| | 2 | 0.064±0.037 | 0.105±0.029 | 0.056±0.028 | 0.073±0.015 | 0.022±0.016 | 0/40 |
| | 4 | **0.030±0.026** | **0.053±0.031** | **0.036±0.013** | 0.026±0.019 | 0.007±0.009 | 0/40 |
| | 6 | 0.032±0.039 | 0.076±0.045 | 0.041±0.017 | **0.010±0.015** | **-0.000±0.000** | 1/40 |
| | MWR+SB | 0.380±0.118 | 0.308±0.092 | 0.456±0.123 | 0.380±0.096 | 0.377±0.107 | 0/40 |
| La11 ($J$=20, $M$=5) | 0 | -0.009±0.046 | **-0.028±0.033** | **-0.026±0.038** | -0.005±0.039 | 0.025±0.051 | 22/40 |
| | 2 | **-0.018±0.045** | -0.009±0.061 | -0.022±0.044 | **-0.020±0.033** | **-0.021±0.033** | **27/40** |
| | 4 | 0.020±0.052 | 0.003±0.036 | 0.019±0.063 | 0.039±0.041 | 0.019±0.056 | 15/40 |
| | 6 | 0.029±0.086 | 0.020±0.103 | 0.036±0.090 | 0.016±0.075 | 0.046±0.068 | 17/40 |
| | MWR+SB | 0.225±0.108 | 0.183±0.123 | 0.172±0.077 | 0.265±0.091 | 0.279±0.089 | 1/40 |

**Instead, it relies on the objective value as the reward in reinforcement learning, requiring the model to independently search for good solutions. This approach may result in an insufficient solution exploration during the training of L2D, likely due to the abundance of local optima in the scheduling problem addressed in this study. For example, the average objective value of L2D for the La16 training dataset is 12.8% worse than that in the solutions of the training dataset (1430.85: L2D, 1268.01: dataset), indicating that L2D could not find better solutions than the solutions in the training dataset.**

By carefully reviewing Table 2, we can see that the neighborhood of the inference of JSPformer is better than that of JSP-DNN. With the La11 dataset, JSP-DNN inferred better solutions than JSPformer, but the opposite result was obtained when incorporating local optimization. This result means that it is difficult to find a better solution in the neighborhood of the inference solution of JSP-DNN, while a better solution can be easily obtained in the neighborhood of the inference solution of JSPformer. The same tendency can be observed for the other datasets; when using JSP-DNN+Opt, the performance is better with a relatively large $K$. This suggests the necessity to search for solutions that are far from the inference of JSP-DNN.

## 5.6 Experimental Results for Job-variant Setting

The experimental results for job-variant settings are shown in Table 3, which lists the gap for each number of removed jobs $L$ in addition to the overall results. As a baseline, we also reported numerical results with MWR and a shifting bottleneck which performed best among classical heuristics in the previous experiment. Comparing Table 3 with Table 2, we can see that JSPformer worked well for training with a variable number of jobs. For the La21 and La11 datasets, the accuracy tended to decrease as the number of removed jobs $L$ increased. Considering the fact that JSPformer worked well for these two datasets in the job-fixed setting, the decrease was caused by the number of instances per set of jobs. The larger $L$ results in more patterns of a set of jobs; for example, 3!=6 patterns of a set of jobs are created from the original instance by removing 3 jobs. In contrast, the gap remained large for the La16 dataset. In this dataset, the number of machines is equal to or less than the number of jobs; therefore, the machines are relatively free when $L$ is large. In such a condition, many schedules are nearly optimal, resulting in a smaller gap when $L$ is large. After local optimization, the effect of $L$ became smaller, which implies that the local optimization efficiently compensates for the small datasets during the training. Particularly in the La21 and La16 datasets, better results were produced with a larger $K$ when using $L = 3$. This suggests that when the number of training data is small, it is effective to use relatively large $K$ for the local optimization to supplement the solution accuracy.

# 6 Discussions

## 6.1 Relations among Optimization Time Limit, Problem Size, and Solution Quality

Figure 6 shows an example of the evolution of the objective value and its lower bound over time while changing the time limit from one minute to five minutes and changing the number of optimization-based jobs $K = |\mathcal{J}_o|$ using an La21 instance. In application, it is reasonable to choose an appropriate $K$ after testing with several instances and checking their objective values and their best bounds. We can see that a too-small $K$ causes the simplified problem (P') to have a worse optimal solution than (P), while a too-large $K$ causes a larger gap to remain between the incumbent (the best solution found) and the best bound. For this instance, the local optimization with $K = 4$ output the best solution of 1,563 compared to other solutions (1,631 for $K = 2$, 1,569 for $K = 3$, and 1,637 for $K = 6$). In general, the size of $K$ affects the computation time exponentially. In our computing environment, however, the objective values after five minutes were not much better than the solutions at one minute; for this reason, we experimented with the computation time set to one minute for the local optimization.

## 6.2 Limitations

We have discussed the advantages of JSP-former for both job-fixed and job-variant settings, but our model also has limitations. We currently recognize three main difficulties: (d1) preparing a dataset with a more accurate solution and more complex instances, (d2) **handling instances with more practical and complex constraints,** and (d3) **creating a general inference model concerning the task assignment.**

The first difficulty derives from the fact that **the JSP is NP-hard and hence the computation time to prepare the exact solution grows exponentially along with the problem size.** To tackle this problem, we need to improve the exact algorithm or reduce the problem size which can be solved within a few hours. This difficulty also implies that it is challenging to make evaluation metrics using optimal solutions. **The second difficulty stems from the first difficulty of creating**

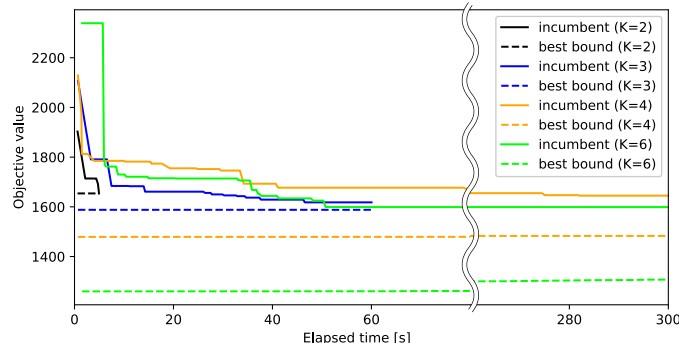

Figure 6: Evolution of objective value and its lower bound over time calculated by CBC while changing the number of optimization-based jobs $K = |\mathcal{J}_o|$. Incumbent refers to the best objective value at the time. Calculation is finished when the time limit is reached or the objective value matches the lower bound.

**datasets; for this problem, a more specialized solver, such as CP-SAT, can be used. However for a broader range of problems with more complex constraints, as used in this paper, a general-purpose solver remains necessary.**

**Regarding the third difficulty, our study focuses on constructing a model tailored to specific situations, rather than addressing unknown task assignments, similar to the existing JSP-DNN. We believe this is reasonable in scenarios where similar instances are repeatedly solved. However, we acknowledge that constructing a general-purpose model capable of handling unknown task assignments is another important research direction. To evaluate the generality of JSPformer, we additionally trained using multiple datasets of different task assignments (see Appendix G). While this experiment demonstrates that our model can be trained across multiple datasets, there remain limitations when dealing with unknown task assignments. Therefore, further research will be needed to efficiently solve instances with unknown task assignments.**

## 7 Conclusion and Future Work

In this paper, we have proposed JSPformer, a data-driven inference model for the **scheduling in the job-variant settings**, and JSPformer+Opt, a hybrid model of JSPformer and local optimization, to solve the **JSPRT** within a few minutes. Our approach has addressed two key challenges: (i1) handling a variable number of jobs and (i2) improving the solution quality within a given time limit using local optimization. JSPformer tackles the first challenge by the first issue by encoding the input data to job-wise features and using Set Transformer, while local optimization manages the second challenge by efficiently adjusting the problem size. We conducted numerical experiments through two settings; namely, job-fixed and job-variant settings. The first experiment in the job-fixed setting show that JSPformer+Opt inferred a better solution than the existing inference-based models. **Furthermore, the second experiment showed the capability of JSPformer(+Opt) to handle job-variant setting.**

**Looking ahead, two key directions for future research stand out. First, it is essential to develop methods that automatically select the optimal approach based on the available time constraints. For example, when sufficient computational time is available, exact algorithms may prove to be more effective. Second, it is crucial to explore how data-driven approaches can be used to address a broader range of optimization problems. In this study, we prepared the dataset using a B&B solver, but the same method can be applied to other mixed-integer optimization problems. Moreover, considering that large language models can now handle discrete and combinatorial structures, machine learning is expected to further improve the efficiency of solving combinatorial optimization problems. In this paper, we adopted an approach that combined partial exact solutions depending on the available computation time. However, an important future research challenge is to develop data-driven methods that can determine the optimal combination of techniques under time constraints. Our findings mark an important first step in this direction, and we believe they will make a meaningful contribution to the ongoing academic development in this field.**

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
