# OpenReview forum: "Inference- and Optimization-based Approximated Solver for Dynamic Job-shop Scheduling Problem"
_TMLR — Rejected by TMLR_

### Review · Reviewer_pP1J · 2024-07-05

**Summary Of Contributions:**

This paper presents JSPformer, a neural network architecture for dynamic job-shop scheduling problems with a variable number of jobs. The key design is to utilize a Set Transformer to encode the input data, so as to handle the job-variant setting. To further improve the solution quality, they present an approach that performs a local optimization to refine the solution generated by the neural network. The results show that JSPformer outperforms both the neural network baseline JSP-DNN and classical heuristics in the job-fixed setting. In the job-variant setting where JSP-DNN is not applicable, JSPformer also outperforms the classic heuristics MWR+SB. In particular, JSPformer + Opt is often able to find better solutions than calling the exact solver to search for solutions within the time limit of 30 mins, while JSPformer + Opt produces solutions within a minute.

**Audience:**

Yes

**Broader Impact Concerns:**

No concerns.

**Claims And Evidence:**

Yes

**Requested Changes:**

1. Please add missing citations and discussion of related work.

2. Evaluate existing neural network baselines for dynamic job scheduling.

3. Explain the intuition why JSPformer outperforms JSP-DNN, and add details on the sizes of JSPformer and JSP-DNN.

4. Present the prediction error and constraint violation measure by exact match. For example, for constraint violation, present the percentage of instances that violate each type of constraints.

5. Add results of training a single model with all data to solve different kinds of dynamic JSP tasks.

**Strengths And Weaknesses:**

Strengths:

1. Solving dynamic job-shop scheduling problems is an important task with a lot of real-world applications. It is also a suitable domain where it is time-consuming for exact solvers to solve large-scale problems, thus learning good heuristics with neural networks is promising.

2. The empirical results are decent.

Weaknesses:

1. The proposed approach lacks novelty. The main design of the neural network architecture is to use the Set Transformer as the encoder. Also, the approach of combining inference-based solutions and optimization-based solutions is straightforward, and it is not surprising that adding the optimization step improves the performance.

2. This work lacks the discussion of related work. There has been a line of work on designing deep learning techniques for job scheduling problems, especially those using deep reinforcement learning [1][2]. The authors did not cite and discuss these works. In particular, there have been lots of works on using deep learning for dynamic job scheduling problems, including [3][4]. However, the authors did not cite and discuss these works, and also did not compare to neural network baselines for dynamic job scheduling.

3. The authors did not explain the intuition why JSPformer outperforms JSP-DNN. Also, the paper lacks details on the sizes of JSPformer and JSP-DNN.

4. It is more informative to measure the prediction error and constraint violation via exact match. For example, for constraint violation, it is better to present the percentage of instances that violate each type of constraints.

5. Have the authors tried training a single model with all data to solve different kinds of dynamic JSP tasks? How do the results compare to training individual models?

[1] Chen et al., Learning to Perform Local Rewriting for Combinatorial Optimization, NeurIPS 2019.
[2] Mao et al., Resource Management with Deep Reinforcement Learning, Proceedings of the 15th ACM Workshop on Hot Topics in Networks, 2016.
[3] Chen et al., An End-to-End Deep Learning Method for Dynamic Job Shop Scheduling Problem.
[4] Song et al., Dynamic Job-Shop Scheduling Based on Transformer and Deep Reinforcement Learning.

---

> ### Author Response · Authors · 2024-09-05
> **Response to the Reviewer pP1J's comments**
>
> Thank you for your valuable comments and suggestions. We acknowledge and appreciate the insightful ideas to analyze our model in detail.
> We revised our manuscript and the revised scripts are highlighted in blue. Below, we address your comments.
>
> > 1. Please add missing citations and discussion of related work.
> > 2. Evaluate existing neural network baselines for dynamic job scheduling.
>
> Thank you for pointing out missing citations. We have added the citations [1]-[4] and the reinforcement learning method L2D [5].
> While a comparison with the baselines [1]-[4] is not appropriate for the following reasons, we instead report the results for [5], a deep learning-based model for job shop scheduling.
>
> [5]: Zhang et al., Learning to Dispatch for Job Shop Scheduling via Deep Reinforcement Learning, NeurIPS 2020.
>
> The first two baselines [1], [2] cannot handle the job-specific task ordering in job-shop scheduling.
> The third baseline [3] focuses on the machine failure as the dynamic feature, rather than on the varying release time.
> The fourth baseline [4] chooses an appropriate dispatching rule among several candidates; however, in our experiment, no dispatching rule performed better than JSPformer or JSP-DNN. In addition, the latter baselines [3], [4], the codes are not available.  Instead, we have evaluated L2D [5], which is another deep reinforcement learning-based model designed to construct an appropriate dispatching rule.
>
> Below, we report the results of L2D for the La21, La16, and La11 datasets, which have been added to Table 2. For fair comparison, we also report L2D+Opt, a hybrid model of L2D and local optimization. We used the same training datasets to train L2D.
> Since the solution quality of L2D is better than the previous result of JSP-DNN, we investigated the reason behind this outcome. The additional analysis indicates that the hyperparameter settings account for the difference. We originally used the dual learning rate $\rho$ in the same range ($10^{-3} \le \rho \le 10^{-2}$) as the JSP-DNN paper, but a smaller $\rho$ actually resulted in better performance. In the table, we updated the results of JSP-DNN and JSPformer with the smaller $\rho$ ($10^{-5} \le \rho \le 10^{-3}$). We updated the hyperparameter settings in Appendix D.

---

> ### Author Response · Authors · 2024-09-05
> **Response to the Reviewer pP1J's comments**
>
> ### Additional experiments with L2D [5] and re-experiments with smaller $\rho \in [10^{-5}, 10^{-3}]$ (best values in bold)
> | La21          | K   | Optimality Gap ↓       | \# Better Sol. ↑ |     | La16          | K   | Optimality Gap ↓      | \# Better Sol. ↑ |     | La11          | K   | Optimality Gap ↓       | \# Better Sol. ↑ |
> | ------------- | --- | ---------------------- | ---------------- | --- | ------------- | --- | --------------------- | ---------------- | --- | ------------- | --- | ---------------------- | ---------------- |
> | JSPformer     | 0   | -0.033 $\pm$ 0.037     | 9/10             |     | JSPformer     | 0   | 0.053 $\pm$ 0.014     | 0/10             |     | JSPformer     | 0   | -0.082 $\pm$ 0.053     | 9/10             |
> | JSPformer+Opt | 2   | -0.067 $\pm$ 0.044     | 9/10             |     | JSPformer+Opt | 2   | 0.037 $\pm$ 0.010     | 0/10             |     | JSPformer+Opt | 2   | **-0.106 $\pm$ 0.036** | 10/10            |
> |               | 3   | **-0.069 $\pm$ 0.036** | 9/10             |     |               | 4   | **0.016 $\pm$ 0.014** | 0/10             |     |               | 4   | -0.094 $\pm$ 0.037     | 10/10            |
> |               | 4   | -0.041 $\pm$ 0.029     | 9/10             |     |               | 6   | 0.049 $\pm$ 0.028     | 0/10             |     |               | 6   | -0.056 $\pm$ 0.056     | 7/10             |
> | JSP-DNN       | 0   | -0.012 $\pm$ 0.026     | 7/10             |     | JSP-DNN       | 0   | 0.077 $\pm$ 0.017     | 0/10             |     | JSP-DNN       | 0   | -0.097 $\pm$ 0.034     | 10/10            |
> | JSP-DNN+Opt   | 2   | -0.058 $\pm$ 0.038     | 9/10             |     | JSP-DNN+Opt   | 2   | 0.071 $\pm$ 0.014     | 0/10             |     | JSP-DNN+Opt   | 2   | -0.102 $\pm$ 0.035     | 10/10            |
> |               | 3   | -0.051 $\pm$ 0.037     | 8/10             |     |               | 4   | 0.059 $\pm$ 0.022     | 1/10             |     |               | 4   | -0.094 $\pm$ 0.032     | 10/10            |
> |               | 4   | -0.024 $\pm$ 0.031     | 7/10             |     |               | 6   | 0.032 $\pm$ 0.014     | 1/10             |     |               | 6   | -0.080 $\pm$ 0.035     | 10/10            |
> | L2D           | 0   | 0.043 $\pm$ 0.053      | 1/10             |     | L2D           | 0   | 0.180 $\pm$ 0.047     | 0/10             |     | L2D           | 0   | -0.058 $\pm$ 0.029     | 10/10            |
> | L2D+Opt       | 2   | 0.008 $\pm$ 0.064      | 6/10             |     | L2D+Opt       | 2   | 0.102 $\pm$ 0.030     | 0/10             |     | L2D+Opt       | 2   | -0.080 $\pm$ 0.029     | 10/10            |
> |               | 3   | -0.028 $\pm$ 0.057     | 9/10             |     |               | 4   | 0.072 $\pm$ 0.032     | 0/10             |     |               | 4   | -0.070 $\pm$ 0.029     | 10/10            |
> |               | 4   | -0.032 $\pm$ 0.035     | 8/10             |     |               | 6   | 0.049 $\pm$ 0.024     | 0/10             |     |               | 6   | -0.041 $\pm$ 0.035     | 9/10             |
>
> Since our results show that JSPformer outperforms L2D across all three datasets, we will keep our claim in the paper.
>
> A possible factor for the difference between L2D and JSPformer could be the use of the training dataset. While JSP-DNN and JSPformer learn from the solutions in the training dataset, L2D does not use these solutions and uses the objective value as the reward in reinforcement learning, requiring the user to search for good solutions on its own. This could result in an insufficient solution exploration during the training of L2D, likely due to the abundance of local optima in the scheduling problem addressed in this study. For example, the average objective value of L2D for the La16 training dataset is 12.8% worse than that in the solutions of the training dataset (1430.85: L2D, 1268.01: dataset), which shows that L2D could not find better solutions than the solutions in the training dataset. We added this discussion to the paper (Section 5.5).
>
> > 3-1. Explain the intuition why JSPformer outperforms JSP-DNN.
>
> JSP-DNN is designed under the assumption that the all jobs are released simultaneously, which results in less accurate results compared to our proposed method. In contrast, JSPformer is designed for scenarios with the different release times by jobs. As mentioned in the response to the next request, JSP-DNN particularly struggles to satisfy both release time constraints and no-overlap constraints simultaneously. We added this intuitive explain to the discussion in Section 5.5.

---

> ### Author Response · Authors · 2024-09-05
> **Response to the Reviewer pP1J's comments**
>
> > 3-2. add details on the sizes of JSPformer and JSP-DNN.
>
> The table below shows the number of parameters of JSPformer and JSP-DNN for the three datasets. For a fair comparison, the size of JSPformer in the job-fixed setting is smaller than the size of JSP-DNN by tuning the output dimension of the job-shared NN $k$ in JSPformer (see Figure 3(b) in our paper). We added this table in Appendix C.
>
> Note that the size of JSP-DNN is fixed in the original paper and the size only depends on the number of jobs and machines.
>
> | # parameters | La21             | La16            | La11            |
> | ------------ | ---------------- | --------------- | --------------- |
> | JSPformer    | 91,914 ($k=128$) | 25,482 ($k=64$) | 23,237 ($k=64$) |
> | JSP-DNN      | 135,450          | 70,900          | 71,900          |
>
> > 4. Present the prediction error and constraint violation measure by exact match. For example, for constraint violation, present the percentage of instances that violate each type of constraints.
>
> For each dataset, we additionally evaluated the prediction error by jobs and the constraint violation by constraint types (c1)-(c3). We report the results below. For the prediction error, we report the average, maximum, and minimum prediction errors per job. For the constraint violation, we present the average number of violated constraints over all constraints, categorized by constraint type.
>
> ### prediction error per job (L2 norm: the mean and standard deviation is computed from 10 instances, best values in bold):
>
> | dataset | model     | average                | max                      | min                    |
> | ------- | --------- | ---------------------- | ------------------------ | ---------------------- |
> | La21    | JSPformer | **476.30 $\pm$ 49.66** | **829.33 $\pm$ 145.85**  | **223.72 $\pm$ 53.19** |
> |         | JSP-DNN   | 750.61 $\pm$ 84.43     | 1215.21 $\pm$ 159.48     | 304.38 $\pm$ 116.56    |
> | La16    | JSPformer | 345.78 $\pm$ 56.34     | 750.64 $\pm$ 218.19      | 136.35 $\pm$ 36.42     |
> |         | JSP-DNN   | **331.80 $\pm$ 64.06** | **709.83 $\pm$ 232.99**  | **131.46 $\pm$ 45.82** |
> | La11    | JSPformer | **512.20 $\pm$ 68.29** | 1085.65 $\pm$ 332.31     | 201.94 $\pm$ 46.74     |
> |         | JSP-DNN   | 543.79 $\pm$ 76.58     | **1074.98 $\pm$ 286.20** | **197.07 $\pm$ 70.30** |
>
> ### average number of violated constraints / number of constraints (best values in bold):
>
> | dataset | model     | task-precedence constraint (c1) | no-overlap constraint (c2) | release-time constraint (c3) |
> | ------- | --------- | ------------------------------- | -------------------------- | ---------------------------- |
> | La21    | JSPformer | **8.59% (11.6 / 135)**          | **3.55% (37.30 / 1050)**   | **6.67% (1.0 / 15)**         |
> |         | JSP-DNN   | 25.48% (34.4 / 135)             | 11.3% (118.3 / 1050)       | **6.67% (1.0 / 15)**         |
> | La16    | JSPformer | **10.78% (9.7 / 90)**           | **4.09% (18.4 / 450)**     | 11.00% (1.1 / 10)            |
> |         | JSP-DNN   | 16.00% (14.4 / 90)              | 10.47% (47.1 / 450)        | **4.00% (0.4 / 10)**         |
> | La11    | JSPformer | 0.25% (0.2 / 80)                | **5.49% (52.2 / 950)**     | 2.00% (0.4 / 20)             |
> |         | JSP-DNN   | **0.13% (0.1 / 80)**            | 8.53% (81.0 / 950)         | **0.00% (0.0 / 20)**         |
>
> The most notable difference was in the number of violated no-overlap constraints, where JSPformer has significantly fewer violations than JSP-DNN. These differences are likely reflected in the optimality gap; generally, when many of the no-overlap constraints are not satisfied, the greedy algorithm may construct solutions that deviate significantly from the inference solution, leading to a decrease in solution quality. In contrast, the prediction error of JSP-DNN was better for two of the three datasets, but it seemed to have a smaller impact on the optimality gap.
>
> We added these results in the appendix F.

---

> ### Author Response · Authors · 2024-09-05
> **Response to the Reviewer pP1J's comments**
>
> > 5. Add results of training a single model with all data to solve different kinds of dynamic JSP tasks.
>
> We appreciate your suggestions for our model. Below, we additionally report the experimental results of JSPformer trained with both the La21 and La16 datasets, referred to as JSPformer-shared. Since the input dimension of Job-shared NN in JSPformer depends on the number of machines (see Figure 3(b) in our paper) and the La11 dataset has a different number of machines, we used the La21 and La16 datasets for this experiment.
> In this experiment, we used larger $k (=256)$ to improve the model capacity, and the number of parameters is 347,658.
>
> ### Additional experiments of JSPformer trained with both the La21 and La16 datasets (best values in bold)
>
> | La21 (J=15, M=10)    | K   | Optimality Gap ↓       | \# Better Sol. ↑ |     | La16 (J=10, M=10)    | K   | Optimality Gap ↓      | \# Better Sol. ↑ |
> | -------------------- | --- | ---------------------- | ---------------- | --- | -------------------- | --- | --------------------- | ---------------- |
> | JSPformer-shared     | 0   | 0.036 $\pm$ 0.042      | 1/10             |     | JSPformer-shared     | 0   | 0.137 $\pm$ 0.034     | 0/10             |
> | JSPformer-shared+Opt | 2   | -0.044 $\pm$ 0.036     | 9/10             |     | JSPformer-shared+Opt | 2   | 0.102 $\pm$ 0.040     | 0/10             |
> |                      | 3   | -0.057 $\pm$ 0.030     | 10/10            |     |                      | 4   | 0.036 $\pm$ 0.014     | 0/10             |
> |                      | 4   | -0.058 $\pm$ 0.032     | 10/10            |     |                      | 6   | 0.042 $\pm$ 0.022     | 1/10             |
> | JSPformer            | 0   | -0.033 $\pm$ 0.037     | 9/10             |     | JSPformer            | 0   | 0.053 $\pm$ 0.014     | 0/10             |
> | JSPformer+Opt        | 2   | -0.067 $\pm$ 0.044     | 9/10             |     | JSPformer+Opt        | 2   | 0.037 $\pm$ 0.010     | 0/10             |
> |                      | 3   | **-0.069 $\pm$ 0.036** | 9/10             |     |                      | 4   | **0.016 $\pm$ 0.014** | 0/10             |
> |                      | 4   | -0.041 $\pm$ 0.029     | 9/10             |     |                      | 6   | 0.049 $\pm$ 0.028     | 0/10             |
>
> These results indicate that JSPformer has the capability to be trained using multiple instances with different jobs. We added these result to the appendix G.

---

### Review · Reviewer_DMH6 · 2024-08-09

**Summary Of Contributions:**

The paper proposes an inference model called JSPFormer for Job-shop Scheduling (JSP) problems with release times and changing number of jobs. JSPFormer uses a set transformer to handle the input of the problem, which is a set of jobs. They also use local optimization  (LO) that improves the quality of the solution obtained by JSPFormer (followed by a feasibility recovery procedure if necessary.) Finally, they compare the performance of JSPFormer with JSP-DNN (an existing baseline method) for problems with a fixed number of jobs and with a heuristic for problems with a changing number of jobs.

**Audience:**

Yes

**Claims And Evidence:**

Yes

**Requested Changes:**

(i) I think it would be better to reposition the paper with respect to the weakness I mentioned above.

(ii) I noticed an error in the mathematical model. Constraint c1 should be written for all tasks of all jobs. Currently, it is written only for all p.

(iii) Also, in Figure 5, both the dashed and solid lines were given the same label (i.e., best bound (K=..)).

**Strengths And Weaknesses:**

*Strengths:*

(i) I think using a set transformer to handle the input is a novel and nice idea since in practice the problem is solved repeatedly with instances by a changing number of jobs. Therefore, JSPFormer provides a solution for more realistic problems.

(ii) Although it is a simple idea, the previous works such as JSP-DNN do not consider using LO for the inferred solution. The results show that LO is useful for all instances.

*Weaknesses:*

(i) As far as I understand, the paper is incorrectly positioned. More specifically, I think the proposed model does not solve a dynamic JSP. The reason is that when the problem is solved, the information about the jobs including the release times is available. Therefore, the problem is solved once (at the beginning of the planning horizon) and for all. Hence, it is static. The problem would be dynamic, for example, if a new job arrives after an initial schedule has been constructed and a re-optimization is needed with the arrival of the new job. The existence of release times does not directly imply that the problem is dynamic.

This is a major weakness and should be carefully addressed or clarified before the potential acceptance of the paper.

---

> ### Author Response · Authors · 2024-09-05
> **Response to the Reviewer DMH6's Comments**
>
> Thank you for your thoughtful comments and suggestions. We are delighted that the novelty of our model.
> We revised our manuscript and the revised scripts are highlighted in blue. Below, we address your comments.
>
> > (i) I think it would be better to reposition the paper with respect to the weakness I mentioned above.
>
> Thank you for your valuable feedback.
> The proposed models JSPformer (+Opt) is designed to solve a JSP with release time constraint at regular intervals, such as every few minutes. Our model can solve JSP in job-variant settings, i.e., varying the set of jobs depending instances.
>
> But as you pointed out, the problem itself is solved as a static one since the problem has all information including the processing time and the release time.
> In response to the comment, we referred to the problem we are addressing as JSPRT (JSP with Release Time) instead of dynamic JSP.
> Also, we avoided stating that JSPformer is a method for solving dynamic JSP. We revised our paper as we focus on the job-variant setting in the introduction and Figure 1.
>
> > (ii) I noticed an error in the mathematical model. Constraint c1 should be written for all tasks of all jobs. Currently, it is written only for all p.
> > (iii) Also, in Figure 5, both the dashed and solid lines were given the same label (i.e., best bound (K=..)).
>
> Thank you for pointing out our typos. We corrected them. In Figure 6 (the index was changed from 5 to 6 since we add one additional figure), the solid line actually represents the incumbent (objective value).

---

### Review · Reviewer_hUm4 · 2024-08-21

**Summary Of Contributions:**

This paper presents a hybrid method combining data-driven AI and optimization for the JSP with release dates. The proposed approach combines a Set Transformer, a feasibility recovery step performed via a greedy algorithm, plus an optional optimization phased based on local search.  Unlike a similar published method, the proposed technique can handle release dates, ad well as instances with a variable number of jobs. The method proves competitive in an experimentation on a custom benchmark built by altering a limited set of standard JSP instances.

**Audience:**

Yes

**Broader Impact Concerns:**

In my opinion, there are not significant ethical concerns for this work

**Claims And Evidence:**

No

**Requested Changes:**

Some of the reported weaknesses are such that acceptance cannot be considered without access to additional information. That said, changes that would result in a stronger paper include:

* Expanding coverage of the SotA, mentioning other hybrid approaches and evaluating more in details method from the CP/SAT literature
* Clarifying the extent of the contribution w.r.t. existing approaches
* Switching to a more significant baseline, or at least providing more details about the chosen baseline
* Extending and re-designing the empirical evaluation, making sure that the instance distribution is not as narrow as it appears to be. Failing that, existing limitation should be clearly stated: for example, there's no way to reliable access whether the proposed method can really handle different machine assignments.

**Strengths And Weaknesses:**

**Strengths**

* Support for variable number of jobs
* Support for release dates

In recent years, the idea of combining ML methods and optimization to target decision-making problems has picked up considerable momentum, and for good reason. In many real world setting, it is common to solve multiple similar problems over time, so that the idea of learning from the past to improve future solution attempts has a lot of appeal.

From this point of view, the ability to deal with problems with multiple size is a significant advantage, as it widens the applicability of the approach and opens up the possibility to train over simple, small, instances and apply the outcome of learning to tougher problems.

Support for constraints with practical relevance, such as release dates, can also be quite useful in a practical setting.

------------------------------------------------------------------------------

**Weaknesses**

* Unclear extent of the contributions
* Limited coverage of related work
* Unclear representativeness of the baselines
* Flaws in the empirical evaluation

Unfortunately, the work also has in my opinion multiple, significant, limitations.

First, the paper builds over a number of existing work: the set transformer from (Lee et. al, 2019), the feasibility recovery algorithm from JSP-DNN, and the local optimization step (which bears similarities with know approaches from the optimization community).

* The set transformer is a key component of the proposed method, but its details are only briefly discussed in section 4.2. As it stands, it seems the authors contribution lies mostly on the representation chosen for the transformer input.
* The feasibility recovering algorithm appears to be almost exactly the same as the one used by the JSP-DNN, which however does not appear to support release date constraints. My impression is that the authors are using the same Mathematical Programming model, and simply adding the missing constraint.
* The local optimization step appears to be a variant of Large Neighborhood Search and match closely the structure of the approaches from [1] and [2], though no related work is mentioned on the topic.

Overall, it would be important to clarify the novelty of the contribution of this paper w.r.t. cited (and omitted) related work. My current impression is that the new contributions are limited to the design of the pipeline and of the input representation.



Second, the section on related work is quite brief. Hybrid method combining optimization and data-driven AI have been an area of quite active research in recent years. I'd recommend checking at least [3] and [4] and references therein.

Additionally, Lazy Clause Generation appraoches (a.k.a. CP-SAT) have been top performers (or among top performers) on scheduling problems for several year, and it was strange to see no mention of them; I'd suggest checking work by Peter J. Stuckey and co-authors on the topic.



Third, it is unclear how much the chosen baseline can be considered representative of the SotA. The authors apparently chose to rely on a Mathematical Programming approach, which for the JSP might not be a particularly good choice due to the low-quality of the bound provided by the LP relaxation of most MP models.

The paper does not provide details on the chosen MP model, which would be necessary to gauge its effectiveness and the tightness of the proposed relaxation. Finally, and less importantly, the chosen solver (CBC) has good performance, but cannot compete with SotA tools such as Gurobi, Xpress, or (in the open source domain) SCIP. Having more details on the baseline, and ideally considering a non-MP baseline, would be important.



Finally, I think there are flaws in the empirical evaluation that limit its reliability. The benchmark instances appear to have been obtained my modifying a set of template instances. While this approach might work fine in principle, it seems the authors chose a very limited set of such templates (just 3 instances) and elected to keep the resource assignment fixed.

This is very concerning, since it means that all instances in the benchmark will have strong correlations in their parameter values, making the distribution little representative of real world use cases. In other words, it's quite possible that the hybrid approach is overfitting w.r.t. more realistic distributions, and there no way assess it with the currently available results.

The fact that instances with variable number of jobs were obtained by removal, rather than addition, further prevents us to assess the ability of the proposed approach to generalize.

------------------------------------------------------------------------------

**Minor problems**

* The naming convention for the target problem (dynamic JSP) is unusual. In the scheduling literature, "dynamic" problems are those where parameters (number of jobs, durations, machine assignment) unpredictably change over time. The problem considered in this paper appears to be a just a deterministic JSP with release dates. Scheduling problems with release dates are well-know in the literature, tough they tend to be considered by CP/LCG approaches more often than via Mathematical Programming.
* There is no mention of the training time, parameters, and algorithm for the proposed approach
* pg 6: J and M are the same in classical JSP (there are as many machines as jobs); if this is not the case here, it's worth briefly mentioning it.
* pg 9: "optimality gap" is a strange term, give that it does not refer to a true optimal solution; that said, the authors explain there's a precedent for that and clarify how the metric is computed.
* It is not fully clear whether JSP-DNN already supports release dates

[1] Godard, Daniel, Philippe Laborie, and Wim Nuijten. "Randomized Large Neighborhood Search for Cumulative Scheduling." ICAPS. Vol. 5. 2005.

[2] Vilím, Petr, Philippe Laborie, and Paul Shaw. "Failure-directed search for constraint-based scheduling." Integration of AI and OR Techniques in Constraint Programming: 12th International Conference, CPAIOR 2015, Barcelona, Spain, May 18-22, 2015, Proceedings 12. Springer International Publishing, 2015.

[3] Florent Teichteil-Königsbuch, Guillaume Povéda, Guillermo González de Garibay Barba, Tim Luchterhand, Sylvie Thiébaux:
Fast and Robust Resource-Constrained Scheduling with Graph Neural Networks. ICAPS 2023: 623-633

[4] Jayanta Mandi, James Kotary, Senne Berden, Maxime Mulamba, Victor Bucarey, Tias Guns, Ferdinando Fioretto:
Decision-Focused Learning: Foundations, State of the Art, Benchmark and Future Opportunities. CoRR abs/2307.13565 (2023)

---

> ### Author Response · Authors · 2024-09-05
> **Response to the Reviewer hUm4's Comments**
>
> Thank you for your constructive feedback and suggestions. We appreciate your thorough review and address your comments below.
> We revised our manuscript and the revised scripts are highlighted in blue.
>
> **weaknesses 1:**
> >First, the paper builds over a number of existing work: the set transformer from (Lee et. al, 2019), the feasibility recovery algorithm from JSP-DNN, and the local optimization step (which bears similarities with known approaches from the optimization community).
> >- The set transformer is a key component of the proposed method, but its details are only briefly discussed in section 4.2. As it stands, it seems the authors contribution lies mostly on the representation chosen for the transformer input.
>
> As you pointed out, the novel component of our method lies in the representation of the data input into the transformer. In the job-variant setting, a key challenge is simultaneously handling two different orders: the machine index and the task assignment. In the job-fixed setting, JSP-DNN implicitly represents the task assignment by constructing a job-wise neural network (Figure 3(a) in the paper). However, this approach is not applicable in job-variant settings where constructing a job-wise neural network is not feasible. To address this, we explicitly represent the task assignment through calculating the earliest possible completion time for each task as the cumulative processing time $\gamma$, and simultaneously representing both orders by arranging them according to the machine index. This explicit handling of task assignment in job-variant settings is the primary novelty of our method.
>
> Additionally, we would like to emphasize that JSPformer(+Opt) succeeds in inferring a high-quality solution in the job-variant setting within a few minutes.
> We have revised the manuscript to clearly distinguish our contributions (in the abstract) from the aspects that build on existing researches (as outlined in Figure 2).
>
> **weaknesses 2:**
> > - The feasibility recovering algorithm appears to be almost exactly the same as the one used by the JSP-DNN, which however does not appear to support release date constraints. My impression is that the authors are using the same Mathematical Programming model, and simply adding the missing constraint.
>
> We have mentioned in the appendix B that lines 4-6 of Algorithm 2 were added to handle the release time constraint. These lines are not included in the original algorithm of JSP-DNN and the original algorithm does not support release time constraints. We emphasized it at the first paragraph in Section 4.3.
>
> **weaknesses 3:**
> > - The local optimization step appears to be a variant of Large Neighborhood Search and match closely the structure of the approaches from \[1] and \[2], though no related work is mentioned on the topic.
>
> Thank you for sharing the related work. We included a discussion of these works in our revised manuscript in the end of Section 3.1.
>
> **weaknesses 4:**
> >Second, the section on related work is quite brief. Hybrid method combining optimization and data-driven AI have been an area of quite active research in recent years. I'd recommend checking at least [3] and [4] and references therein.
>
> Thank you for the recommendation. While these studies are not directly related to the specific problem we are addressing, we included them in the related work (3.2) as the studies about the integration of optimization and AI.
>
> [3] and [4] differ from our problem setting in the following points:
>
> - SERENE [3] does not address job-shop scheduling. Specifically, it does not handle the no-overlap constraint. Additionally, the optimization part in [3] is conducted by selecting from sampled options, which differs from our approach.
> - Decision-focused learning [4] is aimed at predicting uncertain parameters within a problem to obtain robust solutions, rather than using machine learning to solve the optimization problem itself.
>
> Our study is focused on the application of machine learning specifically to solve optimization problems.

---

> ### Author Response · Authors · 2024-09-05
> **Response to the Reviewer hUm4's Comments**
>
> **weaknesses 5:**
> > Additionally, Lazy Clause Generation approaches (a.k.a. CP-SAT) have been top performers (or among top performers) on scheduling problems for several year, and it was strange to see no mention of them; I'd suggest checking work by Peter J. Stuckey and co-authors on the topic.
> > Third, it is unclear how much the chosen baseline can be considered representative of the SotA. The authors apparently chose to rely on a Mathematical Programming approach, which for the JSP might not be a particularly good choice due to the low-quality of the bound provided by the LP relaxation of most MP models.
>
> Thank you for pointing out the related work.
> We have revised the paper to reference CP-SAT in the introduction (second paragraph) and the related work (second paragraph in Section 3.1.).
>
> As you mentioned, we use an MP solver to prepare the dataset. However, our main contribution is to construct a supervised learning model that can handle varying numbers of jobs in the dataset, and we do not claim that our model is a better solver compared to all other solvers, such as CP-SAT. The essential point in the experiment is that JSPformer can be trained in the job-variant setting (the second experiment, Table 3) without sacrificing inference quality compared to other models (the first experiment, Table 2). We have revised the paper to clarify our contributions (abstract, introduction, conclusion) and the interpretation of the experimental results (abstract, Section 5.5 and 5.6).
>
> As a future work, we plan to validate the effectiveness of the proposed framework on different types of problems, as mentioned in the revision (Section 7, the second direction for future work).
>
> **weaknesses 6:**
> >The paper does not provide details on the chosen MP model, which would be necessary to gauge its effectiveness and the tightness of the proposed relaxation.
>
> We used binary variables $x_{pp'}^{jj'}$ to represent the no-overlap constraint (c2): $s_p^j +d_p^j \le s_{p'}^{j′} \textrm{ or } s_p^{j′} +d_{p'}^{j′} \le s_p^{j}$ and we formulated (c2) by the following two inequalities with a large constant $A$:
>
> $$
> \begin{matrix}& s_p^j +d_p^j - s_{p'}^{j′} \le Ax_{pp'}^{jj'} \\\\ & s_{p'}^{j'} +d_{p'}^{j'} - s_p^j \le A(1-x_{pp'}^{jj'}) \end{matrix}
> $$
> We have added it in the appendix F.
>
> **weaknesses 7:**
> > Finally, and less importantly, the chosen solver (CBC) has good performance, but cannot compete with SotA tools such as Gurobi, Xpress, or (in the open source domain) SCIP. Having more details on the baseline, and ideally considering a non-MP baseline, would be important.
>
> As you pointed out, CBC may not be as powerful as some other solvers, especially commercial ones. However, we chose CBC because it is one of the most popular free solvers available. The proposed method can also be trained using solutions provided by other solvers; however, our baseline is not the solver itself but other learning-based models.
> We would like to emphasize that the experiments are fair since the same solver was used consistently across all experiments.
>
> As mentioned in the response to the weaknesses 5, we plan to explore more efficient methods for generating training data as a future work. We have added this discussion in the revised paper.

---

> ### Author Response · Authors · 2024-09-05
> **Response to the Reviewer hUm4's Comments**
>
> **weaknesses 8:**
> >Finally, I think there are flaws in the empirical evaluation that limit its reliability. The benchmark instances appear to have been obtained my modifying a set of template instances. While this approach might work fine in principle, it seems the authors chose a very limited set of such templates (just 3 instances) and elected to keep the resource assignment fixed.
> >This is very concerning, since it means that all instances in the benchmark will have strong correlations in their parameter values, making the distribution little representative of real world use cases. In other words, it's quite possible that the hybrid approach is overfitting w.r.t. more realistic distributions, and there no way assess it with the currently available results.
> >The fact that instances with variable number of jobs were obtained by removal, rather than addition, further prevents us to assess the ability of the proposed approach to generalize.
>
> Considering the application, we believe that it is worthwhile to create a solver that specializes in known task assignments.
> Our model is intended to solve instances similar to those in the dataset quickly, not to solve general instances with respect to the task assignment. We believe the assumption is reasonable in scenarios where similar problems are repeatedly solved. Under this scenario, the experiments in the paper have demonstrated the effectiveness of our model.
>
> As mentioned in the limitations section of the paper, we are aware of the challenges associated with creating the dataset. Therefore, in this study, we conducted experiments based on three types of instances. Developing more efficient methods for dataset creation is one of the future work.
> We stated the above points in the limitations.
>
> **minor 1:**
> >- The naming convention for the target problem (dynamic JSP) is unusual. In the scheduling literature, "dynamic" problems are those where parameters (number of jobs, durations, machine assignment) unpredictably change over time. The problem considered in this paper appears to be a just a deterministic JSP with release dates. Scheduling problems with release dates are well-known in the literature, though they tend to be considered by CP/LCG approaches more often than via Mathematical Programming.
>
> According to this comment and the suggestion by Reviewer DMH6, we referred to the problem we are addressing as JSPRT (JSP with Release Time) instead of dynamic JSP.
> As you pointed out, JSPformer solves the scheduling problem as the deterministic one.
> We use the term “release time” instead of “release date” because we assume that the situation is updated at minute intervals. The term “release date” does not convey the impression of a schedule that operates on a minute-by-minute basis.
>
> **minor 2:**
> >- There is no mention of the training time, parameters, and algorithm for the proposed approach.
>
> We added the table below in the appendix D about the number of parameters and the training time where $k$ refers to the hidden dimension in the transformer (Figure 3(b)).
> In the training procedure, we used Adam.
>
> |           | # parameters     |                 |                 | training time [hh:mm:ss] |         |         |
> | --------- | ---------------- | --------------- | --------------- | -------------------- | ------- | ------- |
> |           | La21             | La16            | La11            | La21                 | La16    | La11    |
> | JSPformer | 91,914 ($k=128$) | 25,482 ($k=64$) | 23,237 ($k=64$) | 1:37:21              | 0:21:49 | 1:25:26 |
> | JSP-DNN   | 154,340          | 70,900          | 71,900          | 1:28:37              | 0:46:42 | 1:18:23 |
>
> **minor 3:**
> >-  pg 6: J and M are the same in classical JSP (there are as many machines as jobs); if this is not the case here, it's worth briefly mentioning it.
>
> JSPLIB includes instances with $J \neq M$, and the size of JSP instance is represented by $J \times M$ in general.
>
> **minor 4:**
> > - pg 9: "optimality gap" is a strange term, give that it does not refer to a true optimal solution; that said, the authors explain there's a precedent for that and clarify how the metric is computed.
>
> According to this comment, we revised it to simply "gap" instead of "optimality gap".
>
> **minor 5:**
> >- It is not fully clear whether JSP-DNN already supports release dates
>
> JSP-DNN originally cannot handle the release time constraint.
> For the experiments, we instead use the proposed input modification, which we mentioned in Section 5.2.
> We have emphasized that JSP-DNN does not support the release time constraint in the paper (Section 4.3).

---

> ### Author Response · Authors · 2024-09-05
> **Response to the Reviewer hUm4's Comments**
>
> **request changes:**
> > Some of the reported weaknesses are such that acceptance cannot be considered without access to additional information. That said, changes that would result in a stronger paper include:
> > - Expanding coverage of the SotA, mentioning other hybrid approaches and evaluating more in details method from the CP/SAT literature
> > - Clarifying the extent of the contribution w.r.t. existing approaches
> > - Switching to a more significant baseline, or at least providing more details about the chosen baseline.
> > - Extending and re-designing the empirical evaluation, making sure that the instance distribution is not as narrow as it appears to be. Failing that, existing limitation should be clearly stated: for example, there's no way to reliable access whether the proposed method can really handle different machine assignments.
>
> As a summary of the above responses, we have addressed the request changes as follows:
> For the first and third requests: as addressed in responses to weaknesses 3-5, we expanded the related work and include discussions on CP-SAT and related studies for integration of optimization and AI. We again emphasize that our main contribution is to construct a supervised learning model for different numbers of jobs in the dataset, as mentioned in responses to weaknesses 5.
> For the second request: we clarified the contributions, as mentioned in response to weakness 1, by highlighting the novelty of input modification and the balance between speed and accuracy in the job-variant setting.
> For the fourth request: as mentioned in response to weakness 8, we plan to add a discussion to the limitations. Again, our model is not intended for generalization; rather, it is designed to solve instances with the specific jobs.

---

### Decision · Action_Editor_wQN3 · 2024-10-04

**Recommendation:** Reject

**Comment:**

All reviewers agree that, despite some points of interest, the paper suffers from weak empirical evaluation.  I encourage the authors to take all reviewer feedback into account, and especially the second point above, when reworking their manuscript.

**Audience:**

The paper tackles a problem at the intersection of operations research and machine learning, and as such it is potentially of interest to a subset of TMLR's readers.

**Claims And Evidence:**

The reviewers have identified several limitations with the empirical evaluation, which weaken the claims made by the authors.

**Resubmission Of Major Revision:**

The authors may consider submitting a major revision at a later time.